# Comparison of journal and top publisher self-citation rates in COVID-19 research

**Alvaro Quincho-Lopez**  *

Unidad de Investigación en Bibliometría, Vicerrectorado de Investigación, Universidad San Ignacio de Loyola, Lima, Peru

* alvaro209.ql@gmail.com

## Abstract

### Introduction

Journal self-citation contributes to the overall citation count of a journal and to some metrics like the impact factor. However, little is known about the extent of journal self-citations in COVID-19 research. This study aimed to determine the journal self-citations in COVID-19 research and to compare them according to the type of publication and publisher.

### Methods

Data in COVID-19 research extracted from the Web of Science Core Collection 2020–2023 was collected and further analyzed with InCites. The journals with the highest self-citation rates and self-citation per publication were identified. Statistical comparisons were made according to the type of publication and publishers, as well as with other major infectious diseases.

### Results

The median self-citation rate was 4.0% (IQR 0–11.7%), and the median journal self-citation rate was 5.9% (IQR 0–12.5%). 1,859 journals (13% of total coverage) had self-citation rates at or above 20%, meaning that more than one in five references are journal self-citations. There was a positive and statistically significant correlation of self-citations with the other indicators, including number of publications, citations, and self-citations per publication (p<0.001). Editorial materials contributed more to journal SC with a median self-citation rate of 5%, which was statistically higher than other documents such as articles, letters or reviews (p<0.001). Among the top twelve publishers, the Multidisciplinary Digital Publishing Institute had a median self-citation rate of 8.33% and was statistically higher than the rest (p<0.001). Self-citation rates for COVID-19 were lower than tuberculosis and HIV/AIDS, but self-citations per publication of these diseases were statistically lower than those for COVID-19 (p<0.001).

### Conclusion

Some journals from the Web of Science Core Collection displayed exorbitant journal self-citation patterns during the period 2020–2023. Approximately, one in every five paper

**Data Availability Statement:** All relevant data are within the manuscript and its Supporting Information files.

**Funding:** The author(s) received no specific funding for this work.

**Competing interests:** The authors have declared that no competing interests exist.

references in COVID-19 is a journal self-citation. Types of publication such as editorials engage in this practice more frequently than others, suggesting that in COVID-19 research, self-citing non-citable items could potentially contribute to inflate journal impact factors during the pandemic.

## Introduction

Faced with the threat and uncertainty generated by the Coronavirus Disease 2019 (COVID-19), the scientific community responded quickly by generating new publications at an unprecedented level. Unsurprisingly, those publications have been cited by multiple other papers at incredibly high rates while the pandemic continues to be in the spotlight [1]. In fact, a fifth of the citations received to papers published in 2020 to 2021 were to COVID-19–related papers [2]. Subsequently, the citations received by said articles would play a very important role in the next metrics of different scoring systems to quantify journal's impact [1].

Each year editors of journals enthusiastically await the release of journals' metrics like the impact factor (IF) by the Journal Citation Reports (JCR). Some studies have revealed that journals that published more COVID-19-related articles experienced higher IF growth [3, 4]. Although, in all disciplines, journals experienced a higher IF increase during the period 2020–21 than previous years, this was much more noticeable in those journals with lower IF [4, 5]. Indeed, the massive covidization of research citations has distorted the metrics of scientific journals. However, the demanding race to publish was supported by both the authors who had their manuscripts published in a short time, as well as the journals and the editors who were eager to boost their metrics [2, 6].

Journal self-citations (SCs) occurs when a journal is cited by its own publications. Although SCs can occur regularly based on the relevance of a reference according to the author's own criteria or in the belief that this would improve the probability of successfully passing the journal's peer-review, there are also cases of coercive SC made by the editorial staff during the peer review process usually after a paper has been conditionally accepted for publication in the journal [7, 8]. Previous research raised concerns about the role of "forced" SC in the annual scientometric indicators of the journals or their impact on the position of the journal quartiles [8–11]. This takes on greater relevance in the context of COVID-19, where the race to publish eventually compromised the quality of articles and regular processes under express peer review [12, 13]. Previously, some studies have focused on how COVID-19 has impacted the IF due to the volume of papers and citations without considering self-citations as a mechanism to inflate metrics at the journal level [4, 5], or studies that have characterized the SCs of journals belonging to the JCR in general terms [14, 15].

Several studies have explored the journal SCs in different research areas. Although they have some similarities such as data sources, most frequently the JCR, they also have important differences in the methodology such as the sample size and sample selection. For example, there are some studies that selected for convenience certain specific journals [16–22], while others selected all journals under a JCR category as well as others that restricted it only to journals with an IF [23–26]. There are studies that analyzed results in a single year while others covered several periods [16, 19–22, 27–32]. The summary measure was also heterogeneous between studies. The most frequent was the mean SC rate, such as in the case of sports sciences and ecological sciences, of 6.7% and 11.8%, respectively [27, 28]; while in others a range of values was shown such as emergency medicine (4.8 to 8.1%), plastic and reconstructive surgery (1

to 4.5%), otolaryngology (4 to 11.9%) and anesthesia (4 to 30%) [17, 19, 21, 22]. In other areas such as soil sciences (10%), pediatrics (9%), dermatology (10.5%) and ophthalmology (11.6%) the median SC rate was reported [18, 33–35]. On the other hand, some articles focused on the impact of the journal SC on the IF, for example Sanfilippo et al. and the so-called "journal self-citation (JSC) rate", whose calculation is detailed as the IF including SC minus IF without SC divided by IF including SC, reported similar values in areas such as anesthesiology (4.8%), respiratory system (5.5%), general surgery (6.5%) and critical care (8.8%) [23–26]. Similarly, Opthof found that the % inflation IF by journal SC was 17% in a sample of the top 50 journals under the WoS category 'Cardiac and Cardiovascular Systems', even being higher than 33% in 8 journals [36]. Other relevant results can be found in Table 1.

However, up to now, there are no studies that focus on journal SCs in COVID-19 research, despite the great impact it has had on science and the scientific community, considering its short-term evolutionary nature. Therefore, this study aimed to determine the journal self-citation rates (SCRs) for COVID-19 research in the Web of Science Core Collection (WoSCC), and to identify the journals with the highest SCRs and SC per publication. Additionally, it sought to analyze how SCs rates vary across different types of publications and publishers, and to compare these rates with those in other major infectious disease research. Concurrently, the study evaluated the correlation between SCs and other bibliometric indicators.

## Methods

The WoSCC was used to retrieve all relevant documents in the present study on February 24th, 2024. The lack of self-citation analysis tools and lower availability of bibliometric indicators were the main reasons for not using PubMed or Scopus, respectively [37, 38]. Furthermore, there is a large overlap of journals between these databases, which would generate a large number of duplicates. Additionally, there are specific journal impact indicators for each database, therefore journals from one source cannot be judged using metrics from other sources [37, 39]. Different search terms were extracted from the Medical Subject Headings from PubMed and Emtree from Embase, and from a combination of both a complex search strategy was generated. Because only studies relevant to COVID-19 were sought to be recovered, the search was restricted to the title and keywords, but not the abstract. Then, the results were refined using the available WoSCC filters: (1) document type: exclude all types except for article, letter, review article, editorial material and early access; (2) year of publication: only results from 2020 to 2023 were included. The detailed search strategy can be found in the supporting information (S1 File).

The information collected from WoSCC was exported to InCites (from Clarivate) for further analysis. In this study, SCs were defined as instances where COVID-19 papers cite any previous work from the same journal, regardless of whether those cited papers are about COVID-19 or other topics. Three datasets were analyzed with information regarding COVID-19 (see S2 File): (i) the journal dataset with all journals publishing in COVID-19; (ii) the type of publications dataset; and (iii) the publisher dataset with the top twelve publishers with the highest number of publications on the topic. For the first dataset, the following information was collected from each journal: number of publications, citations, citations without SC, journal IF, and IF without SC, % documents cited, the Category Normalized Citation Impact (CNCI), the Journal Normalized Citation Impact (JNCI), immediacy index, % international collaboration, and the percentage of documents in the top 1% and 10% of highly cited papers in the WoSCC. SC were obtained indirectly by subtracting citations without SC from total citations. The SCR represents the proportion of a journal's SCs to the total citations received multiplied by 100. The same reasoning was followed to calculate the JSC rate, which is detailed in

**Table 1. Brief review of the key literature.**

| Author | Research area | Journals | Source | Period | Main finding | Other findings |
|---|---|---|---|---|---|---|
| Bennet 2024 | Sport sciences | 87 journals in the category of "sport sciences" | JCR | 2013–2022 | Mean SC rate 6.7% | SC rates have increased over time by approximately 10% per year |
| Deora 2023 | Neurosurgery | 7 specific journals | WoS | 2021–2020 | - | There was an increase of journal SC as the years progressed and as the number of articles published by any journal increased |
| Blackledge 2022 | Obstetrics and gynecology (OB/GYN) | 89 OB/GYN journals | JCR | 2010–2019 | - | The number of citable items decreased for top-tertile journals and increased for bottom- and middle-tertile journals |
| Sanfilippo 2022 | Respiratory system | 56 "Respiratory System" journals with an IF | JCR | 2021 | J-SC rate* 5.5% | The J-SC rate was not different according to their focus of interest (multidisciplinary vs. specific, p = 0.28) and publishing options (fully open-access vs. traditional, p = 0.38) |
| Sri-ganeshan 2021 | Emergency | 6 specific journals | JCR | 2020 | mean SC rate 4.8 to 8.1% | Non-significant positive correlation between IF and SC rates |
| Sanfilippo 2021 | General surgery | 85 "surgery" journals with an IF | JCR | 2021 | J-SC rate* 6.5% | The J-SC was significantly higher (p = 0.03), in multidisciplinary journals, followed by "topic-specific" and "broad-interest" (11.5, 8.9, and 5.5%, respectively) |
| Sanfilippo 2021 | Critical care | 35 "critical care medicine" journals with an IF | JCR | 2020 | J-SC rate* 8.8% | The IF was not different between journals according to sub-categories (p = 0.35): "topic-specific", "broad interest", and "multidisciplinary" (2.4, 2.5, and 3.0%, respectively) |
| Ma 2021 | Soil science | 9 specific journals | Scopus | 2018 | median rate SC 10% | There is a trend of an increase in the proportion of SC with the number of authors and references |
| Sanfilippo 2021 | Anesthesiology | 32 anesthesiology journals with an IF | JCR | 2020 | J-SC rate* 4.8% | A higher J-SC rate in specific vs. broad scope anesthesiology journals (25.5% vs. 7.2%) |
| Sundaram 2019 | Orthopedic | 47 orthopedic journals | JCR and SCImago | 1997 to 2017 | - | General-interest orthopedic journals had lower median SC rates compared to specialized journals (6.36% vs. 11.85%, p<0.001) |
| Livas 2018 | Dentistry | 85 "dentistry, oral surgery, and medicine" journals | JCR | 2014 to 2016 | median SC rate per year 10.6 to 13.7% | Open-access journals tended to present lower SCR compared to subscription-based journals |
| Miyamoto 2017 | Plastic and reconstructive surgery | 16 specific journals | JCR | 2009 to 2015 | mean SC rate 1 to 4.5% | Spearman's rank correlation showed that the IF was significantly correlated with the SC rate (r = 0.824, p < 0.001) |
| Mimouni 2016 | Pediatrics | 117 pediatric journals | JCR | 2013 | median SC rate 9% | Spearman's ranked correlation showed that IF was significantly and inversely correlated with SC rate (r = -0.28, P = 0.002; R2 = 0.08). Subspecialty and general pediatrics journal did not differ in terms of SC rate |
| Reiter 2016 | Dermatology | 59 dermatology journals | JCR | 2014 | median SC rate 10.5% | The SC rate was significantly and inversely correlated with the IF (r = 0.23, p = 0.04). Subspecialty journals have a higher SC rate than general dermatology journals |
| Mimouni 2014 | Ophthalmology | 58 ophthalmology journals | JCR | 2013 | median SC rate 11.6% | Spearman's rank correlation showed that the number of SC significantly correlated with the number of publications (R2 = 86.3, p = 0<001). Subspecialty journals had higher SC rate than general journals (p = 0.017) |
| Opthof 2013 | Cardiovascular | top 50 journals under the WoS category 'Cardiac and Cardiovascular Systems' | JCR | 2010 | % inflation IF** 17% | There are 8 journals with inflation scores above one third (33 %) |
| Kurmis 2010 | Medical imaging | 3 specific journals | PubMed | 2004 to 2005 | mean SC rate per article 4.7 to 17.1% | The journal IF was proportional to their % SC |
| Landoni 2010 | Anesthesia and Critical care | 41 anesthesia and critical care journals with an IF | JCR | 1999 to 2009 | median SC rate per year 6.6 to 44.4% | The median SC rate increased every year |

*(Continued)*

**Table 1.** (Continued)

| Author | Research area | Journals | Source | Period | Main finding | Other findings |
|--------|---------------|----------|--------|--------|--------------|----------------|
| Krauss 2007 | Ecological science | 107 ecological journals | JCR | 1998 to 2004 | mean SC rate 11.8% | The total SC rate was negatively correlated with the journal IF (n = 107; R2 = 8.38%; p = 0.003) |
| Motamed 2002 | Otolaryngology | 6 specific journals | WoS | 1997 to 1998 | mean SC rate 4 to 11.9% | No significant correlation was found between SC rates and IF for the 6 journals (r = - 0.3143, p = 0.56) |
| Fassoulaki 2000 | Anesthesia | 6 specific journals | JCR | 1995 to 1996 | mean SC rate 4 to 30% | The citations each journal gave to other journals, including itself, and the citations each journal received from the other journals differed significantly among the six journals (p<0.0001) |

Notes.

*J-SC rate calculation: IF including self-citations minus IF without self-citations divided by IF including self-citations times 100

**% inflation IF calculation: IF including self-citations minus IF without self-citations divided by IF without self-citations times 100

another publication [26], but instead uses the IF with and without journal SC. Additionally, for a small sample of those journals that have more than 600 publications in COVID-19 papers, the number of categories assigned by the JCR and whether they were fully open access (OA) (non-hybrid nor subscription) were manually collected. The Mann–Whitney U-test for unrelated samples was performed separating journals according to their fully OA status and number of categories (1 vs. >1) to find differences according to the SCR and JSC rate.

With the same dataset, a correlation matrix was calculated and graphed based on the Spearman's rank correlation coefficient to correlate the journal SC as an absolute value and in percentage with the previously mentioned indicators. Also, the same dataset of journals was filtered (> 100 publications and > 100 citations) to obtain those journals with the highest SCR and SC per publication values, and graph them as lollipop charts. The main journals were categorized into quartiles according to the JCR. When there were several quartiles according to the edition and/or categories, the best quartile was chosen. When a JCR quartile was unavailable, the Journal Citation Indicator (JCI) quartiles were used, since both metrics have a high correlation [40].

With the publisher dataset, violin plots were used to compare the SCR between the top 12 publishers with the highest number of publications in COVID-19. For this purpose, the *ggstatsplot* library was used [41], which allows comparing the medians using the Kruskal-Wallis test, also indicating the adjusted p-value after Holm's correction as a post hoc test to conduct pairwise comparisons and identify specific group differences. A similar procedure was followed for the type of publication dataset comparing the SCR among articles, reviews, letters, and editorials. Because only up to 150,000 documents could be exported to InCites, the number of documents under the article category was sorted in decreasing order of citations while the content of the other publication types could be completely exported.

To compare the number of SCs for COVID-19 journals, I performed searches for other major infectious diseases such as HIV/AIDS and tuberculosis in the WoSCC (see S3 File). To ensure comparability, the same restrictions were used as for the COVID-19 search strategy, considering the type of publications and years (2020–2023). Nonetheless, in order to avoid any impact of the COVID-19 pandemic on publication dynamics, it was also analyzed in the period 2016–2023. Also, the Kruskal-Wallis test was applied between SC/publication of COVID-19 with the other diseases in both periods 2020–2023 and 2016–2023. All data analyses were conducted on RStudio software version 4.3.0 (Boston, MA, USA). A p-value less than 0.05 was considered statistically significant for all analyses. A flow chart of the search is shown in Fig 1.

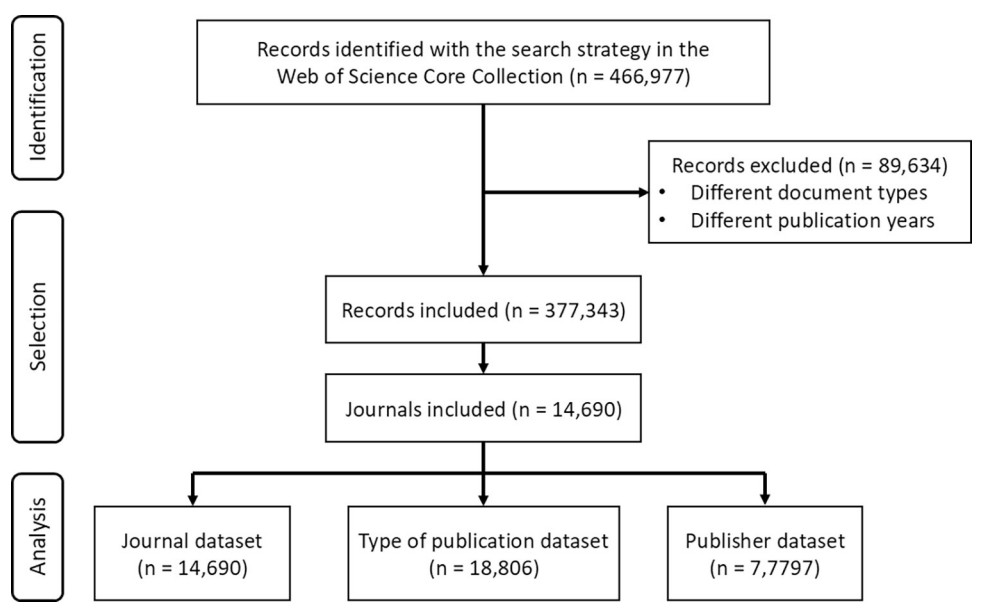

**Fig 1. Flow chart of the data retrieval, selection and analysis.**

## Results

From more than 377,000 documents retrieved in the WoSCC, information of a total 14,690 journals was retrieved. Table 2 corresponds to the most productive journals that have more than 600 publications in the topic. In total, these 45 journals had 69,270 publications in COVID-19, obtaining 1,442,563 citations, of which 77,368 (5.36%) were SCs. Although it could be expected that the most productive journals would be those with the greatest number of citations, this was not the case, with the most cited within Table 2 being very high impact journals such as the *NEJM*, *The Lancet* and *JAMA*, all of which exceeded 100,000 citations. The largest number of SCs were for *International Journal of Environmental Research and Public Health—IJERPH* (MDPI), *Vaccines* (MDPI) and *Science of the Total Environment* (Elsevier). Journals that had a high SCR (>20%) were *Sustainability* (MDPI), *Vaccines* (MDPI), and *Journal of Family Medicine and Primary Care* (Wolters Kluwer). *Science of the Total Environment* (Elsevier) was the journal that had the highest number of SCs per publication (6 SCs), followed by other two journals, one from MDPI and Taylor & Francis (3 SCs, each). When calculating the JSC rate, three MDPI journals lead the indicator, namely *Sustainability*, *Vaccines* and *Nutrients*. No differences were found between the medians of fully OA journals (6.7% [3.6–9.1%]) and hybrid/subscription-based journals (3.9% [1.7–6.6%], p = 0.17), with respect to SCR. Neither for the JSC rate: 6.1% (3.4–8.5%) vs. 3.2% (1.6–7%), respectively (p = 0.14). Nonetheless, journals belonging to 2 or more JCR categories have higher SCR (7.7% [6.1–10.1%] vs. 4.7% [3.2–7.7%]; p = 0.04) and JSC rates (7.4% [5.4–8.8%] vs. 4.4 [2.4–8.3%]; p = 0.04) than those who belong to a specific category. For the full table, see the S2 File.

Based on the full dataset, the median SCR was 4.0% (IQR 0–11.7%), and the median JSC rate was 5.9% (IQR 0–12.5%). 10,822 journals (74% of total coverage) had SCR below 20%, whereas 1,859 journals (13% of total coverage) had SCR at or above 20%, meaning that more than one in five references are journal SC. The Spearman correlation matrix (Fig 2) shows that there was a positive and statistically significant correlation of SC with the other indicators, which means that when the SC increased, these indicators also increased (p<0.001). However, the degree of correlation varied from weak (% documents cited, % international collaboration)

**Table 2. Most productive journals with at least 600 COVID-19 publications indexed in the Web of Science 2020–2023.**

| # | Journal | Publisher | Fully open access/N° categories | Publications | Citations | Citations (excl. SC) | SC | SC% | SC/ Publication | IF 2022 | IF (excl. SC) | JSC rate % |
|---|---------|-----------|-------------------------------|--------------|-----------|---------------------|-----|-----|-----------------|---------|---------------|-----------|
| 1 | International Journal of Environmental Research and Public Health (IJERPH) | MDPI | Yes/3 | 5,823 | 88,511 | 78,485 | 10,026 | 11.33 | 1.72 | n/a | n/a | n/a |
| 2 | PLoS One | PLOS | Yes/1 | 5,033 | 75,717 | 71,764 | 3,953 | 5.22 | 0.79 | 3.7 | 3.5 | 5.41 |
| 3 | Frontiers in Public Health | Frontiers | Yes/2 | 3,607 | 26,269 | 24,127 | 2,142 | 8.15 | 0.59 | 5.2 | 4.8 | 7.69 |
| 4 | Cureus Journal of Medical Science | Cureus→Springer Nature | Yes/1 | 3,518 | 17,123 | 15,885 | 1,238 | 7.23 | 0.35 | 1.2 | 1.1 | 8.33 |
| 5 | Vaccines | MDPI | Yes/2 | 2,919 | 32,266 | 25,384 | 6,882 | 21.33 | 2.36 | 7.8 | 6.5 | 16.67 |
| 6 | Scientific Reports | Nature | Yes/1 | 2,874 | 38,037 | 36,255 | 1,782 | 4.68 | 0.62 | 4.6 | 4.4 | 4.35 |
| 7 | Sustainability | MDPI | Yes/4 | 2,643 | 27,283 | 21,450 | 5,833 | 21.38 | 2.21 | 3.9 | 3.1 | 20.51 |
| 8 | Frontiers in Psychology | Frontiers | Yes/1 | 2,478 | 31,190 | 27,688 | 3,502 | 11.23 | 1.41 | 3.8 | 3.3 | 13.16 |
| 9 | Frontiers in Immunology | Frontiers | Yes/1 | 2,290 | 35,587 | 32,130 | 3,457 | 9.71 | 1.51 | 7.3 | 6.5 | 10.96 |
| 10 | Journal of Medical Virology | Wiley | No/1 | 2,161 | 60,060 | 57,698 | 2,362 | 3.93 | 1.09 | 12.7 | 12.4 | 2.36 |
| 11 | Journal of Clinical Medicine | MDPI | Yes/1 | 2,024 | 24,210 | 22,757 | 1,453 | 6.00 | 0.72 | 3.9 | 3.6 | 7.69 |
| 12 | Viruses | MDPI | Yes/1 | 1,908 | 22,169 | 20,463 | 1,706 | 7.70 | 0.89 | 4.7 | 4.1 | 12.77 |
| 13 | BMJ Open | BMJ | Yes/1 | 1,844 | 17,844 | 17,177 | 667 | 3.74 | 0.36 | 2.9 | 2.8 | 3.45 |
| 14 | Frontiers in Medicine | Frontiers | Yes/1 | 1,645 | 14,973 | 14,450 | 523 | 3.49 | 0.32 | 3.9 | 3.7 | 5.13 |
| 15 | Clinical Infectious Diseases | Oxford UP | No/3 | 1,388 | 54,589 | 53,178 | 1,411 | 2.58 | 1.02 | 11.8 | 11.5 | 2.54 |
| 16 | International Journal of Infectious Diseases | Elsevier | Yes/1 | 1,386 | 41,609 | 40,797 | 812 | 1.95 | 0.59 | 8.4 | 8.2 | 2.38 |
| 17 | Healthcare | MDPI | Yes/2 | 1,332 | 7,652 | 6,994 | 658 | 8.60 | 0.49 | 2.8 | 2.6 | 7.14 |
| 18 | BMC Public Health | Springer Nature | Yes/1 | 1,316 | 13,955 | 13,359 | 596 | 4.27 | 0.45 | 4.5 | 4.3 | 4.44 |
| 19 | Frontiers in Psychiatry | Frontiers | Yes/2 | 1,304 | 17,613 | 16,338 | 1,275 | 7.24 | 0.98 | 4.7 | 4.3 | 8.51 |
| 20 | The BMJ British Medical Journal | BMJ | Yes/1 | 1,265 | 49,353 | 47,753 | 1,600 | 3.24 | 1.26 | 107.7 | 105.8 | 1.76 |
| 21 | International Journal of Molecular Sciences | MDPI | Yes/2 | 1,199 | 13,680 | 12,361 | 1,319 | 9.64 | 1.10 | 5.6 | 5.0 | 10.71 |
| 22 | Heliyon | Elsevier | Yes/1 | 1,123 | 8,119 | 7,657 | 462 | 5.69 | 0.41 | 4.0 | 3.9 | 2.50 |
| 23 | JAMA Network Open | AMA | Yes/1 | 1,116 | 38,232 | 37,224 | 1,008 | 2.64 | 0.90 | 13.8 | 13.4 | 2.90 |
| 24 | Vaccine | Elsevier | No/2 | 1,055 | 13,834 | 12,986 | 848 | 6.13 | 0.80 | 5.5 | 5.2 | 5.45 |
| 25 | Journal of Infection | Elsevier | No/1 | 991 | 29,374 | 28,359 | 1,015 | 3.46 | 1.02 | 28.2 | 27.3 | 3.19 |
| 26 | Science of the Total Environment | Elsevier | No/1 | 972 | 52,992 | 46,971 | 6,021 | 11.36 | 6.19 | 9.8 | 8.8 | 10.20 |
| 27 | The Lancet | Elsevier | No/1 | 939 | 124,859 | 123,750 | 1,109 | 0.89 | 1.18 | 168.9 | 167.8 | 0.65 |
| 28 | Nature Communications | Nature | Yes/1 | 887 | 48,132 | 47,030 | 1,102 | 2.29 | 1.24 | 16.6 | 16.2 | 2.41 |
| 29 | BMC Infectious Diseases | Springer Nature | Yes/1 | 863 | 9,047 | 8,836 | 211 | 2.33 | 0.24 | 3.7 | 3.7 | 0.00 |
| 30 | Medicine | Lippincott | Yes/1 | 813 | 4,248 | 4,101 | 147 | 3.46 | 0.18 | 1.6 | 1.6 | 0.00 |
| 31 | Journal of Medical Internet Research | JMIR | Yes/2 | 809 | 22,738 | 21,346 | 1,392 | 6.12 | 1.72 | 7.4 | 7.0 | 5.41 |
| 32 | New England Journal of Medicine (NEJM) | MMS | No/1 | 791 | 178,317 | 177,243 | 1,074 | 0.60 | 1.36 | 158.5 | 157.6 | 0.57 |
| 33 | Diagnostics | MDPI | Yes/1 | 757 | 5,984 | 5,429 | 555 | 9.27 | 0.73 | 3.6 | 3.3 | 8.33 |
| 34 | Human Vaccines & Immunotherapeutics | Taylor & Francis | Yes/2 | 738 | 8,442 | 7,761 | 681 | 8.07 | 0.92 | 4.8 | 4.4 | 8.33 |
| 35 | Disaster Medicine and Public Health Preparedness | Cambridge UP | No/2 | 724 | 4,196 | 3,929 | 267 | 6.36 | 0.37 | 2.7 | 2.5 | 7.41 |
| 36 | Journal of Family Medicine and Primary Care | Wolters Kluwer | Yes/1 | 722 | 1,773 | 1,411 | 362 | 20.42 | 0.50 | 1.4 | 1.3 | 7.14 |
| 37 | Nutrients | MDPI | Yes/1 | 693 | 17,304 | 14,802 | 2,502 | 14.46 | 3.61 | 5.9 | 5.0 | 15.25 |

*(Continued)*

**Table 2.** (Continued)

| # | Journal | Publisher | Fully open access/N° categories | Publications | Citations | Citations (excl. SC) | SC | SC% | SC/ Publication | IF 2022 | IF (excl. SC) | JSC rate % |
|---|---------|-----------|----------------------------------|--------------|-----------|----------------------|-----|------|-----------------|---------|---------------|------------|
| 38 | Journal of Biomolecules Structure Dynamics | Taylor & Francis | No/2 | 692 | 14,239 | 11,871 | 2,368 | 16.63 | 3.42 | 4.4 | 4.0 | 9.09 |
| 39 | Infection Control and Hospital Epidemiology | Cambridge UP | No/2 | 680 | 6,848 | 6,375 | 473 | 6.91 | 0.70 | 4.5 | 4.2 | 6.67 |
| 40 | Open Forum Infectious Diseases | Oxford UP | Yes/3 | 675 | 6,151 | 5,998 | 153 | 2.49 | 0.23 | 4.2 | 4.0 | 4.76 |
| 41 | Annals of Medicine and Surgery | Lippincott | Yes/1 | 673 | 3,581 | 3,311 | 270 | 7.54 | 0.40 | 1.7 | 1.6 | 5.88 |
| 42 | European Review for Medical and Pharmacological Sciences | Verduci | Yes/1 | 672 | 7,142 | 6,563 | 579 | 8.11 | 0.86 | 3.3 | 3.1 | 6.06 |
| 43 | Journal of the American Medical Association (JAMA) | AMA | No/1 | 650 | 105,538 | 104,752 | 786 | 0.74 | 1.21 | 120.7 | 119.6 | 0.91 |
| 44 | Journal of Infectious Diseases | Oxford UP | Yes/3 | 646 | 13,794 | 13,438 | 356 | 2.58 | 0.55 | 6.4 | 6.2 | 3.13 |
| 45 | Frontiers in Pharmacology | Frontiers | Yes/1 | 632 | 7,989 | 7,559 | 430 | 5.38 | 0.68 | 5.6 | 5.1 | 8.93 |

Notes. AMA: American Medical Association; BMC: BioMed Central; JMIR: Journal of Medical Internet Research; MDPI: Multidisciplinary Digital Publishing Institute; MMS: Massachusetts Medical Society; OUP: P Oxford University Press; PLOS: Public Library of Science; UP: University Press.

to moderate (IF, CNCI, JNCI, % documents in the top 1% and 10% of highly cited papers) in some cases, and even strong in others (publications, citations, and SC per publication).

To avoid exaggerated values of SCR and SC per publication in journals with few citations and/or publications, the journal dataset was filtered to only journals with at least 100 publications and citations, leaving 645 journals for analysis. Fig 3A shows the journals with the highest SCR, led by *Bali Medical Journal* and *Orvosi Hetilap* with more than 50%, followed by two others with >40%, five with >30% and eighteen with >21%. Also, Fig 3B shows that *International Journal of Contemporary Hospitality Management* had the highest number of SCs per publication (8 SC for each publication), followed by *Science of Total Environment* with 6 SC/ publication.

When the publication type dataset was analyzed, it was observed that the editorial materials contributed more to journal SC with a median SCR of 5%, which was statistically higher than other documents such as articles, letters or reviews (p<0.001) (see Fig 4).

Fig 5 corresponds to the 12 most productive publishers in COVID-19 research distributed according to the level of journal SC (%). Fig 5A shows the 6 most productive publishers, led by Elsevier (1,648 journals) followed by Springer Nature (1,540 journals). However, MDPI was the one with the highest median SCR with 8.33% and was statistically higher than the other five publishers (p<0.001). Taylor & Francis also stands out in this group for its SCR (4.76%). Fig 5B shows the following most productive publishers, where SAGE stands out for its SCR (4.76%). Nature (0.97%) is the publisher with the lowest SCR, whose results were statistically lower than the other publishers, although no significant statistical differences were found when compared with PLOS (1.85%).

As shown in Table 3, the number of publications, journals and especially citations relevant to COVID-19 research exceeds that of other infectious diseases even in the last 8 years. As expected, SCs in COVID-19 were also much higher than the SCs generated in tuberculosis and HIV/AIDS research. When comparing the SCR, tuberculosis and especially HIV/AIDS research take the lead. By adjusting the number of SCs per publication, it was obtained that the

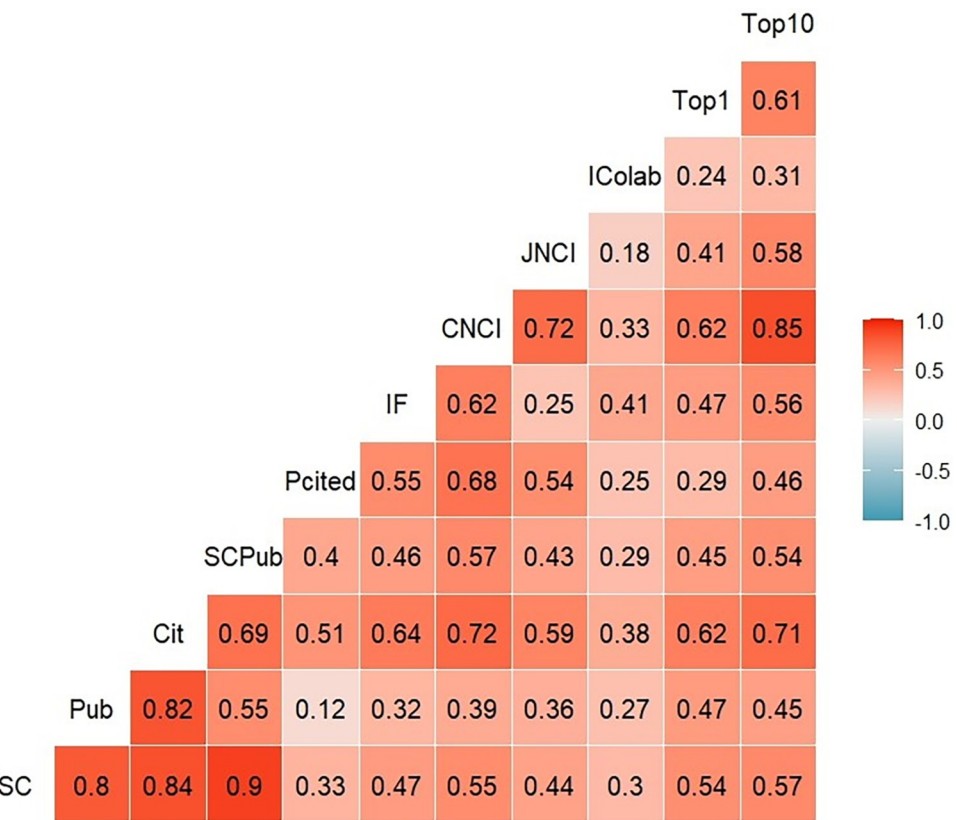

**Fig 2. Correlation matrix of journal self-citations in COVID-19 research with other indicators.** Note. SC, self-citation; Pub, publication; Cit, citations; SCPub, self-citation per publication; Pcited, % documents cited; IF, impact factor; CNCI, Category Normalized Citation Impact; JNCI, Journal Normalized Citation Impact; Top1, % documents in the top 1%; Top10, % documents in the top 10% of highly cited papers.

result for COVID-19 was higher than other infectious diseases in the same timespan (2020–2023). Even when increasing the timespan to the last 8 years (2016–2023), COVID-19 SC per publication still remained higher than the others. In both periods, there was a significant

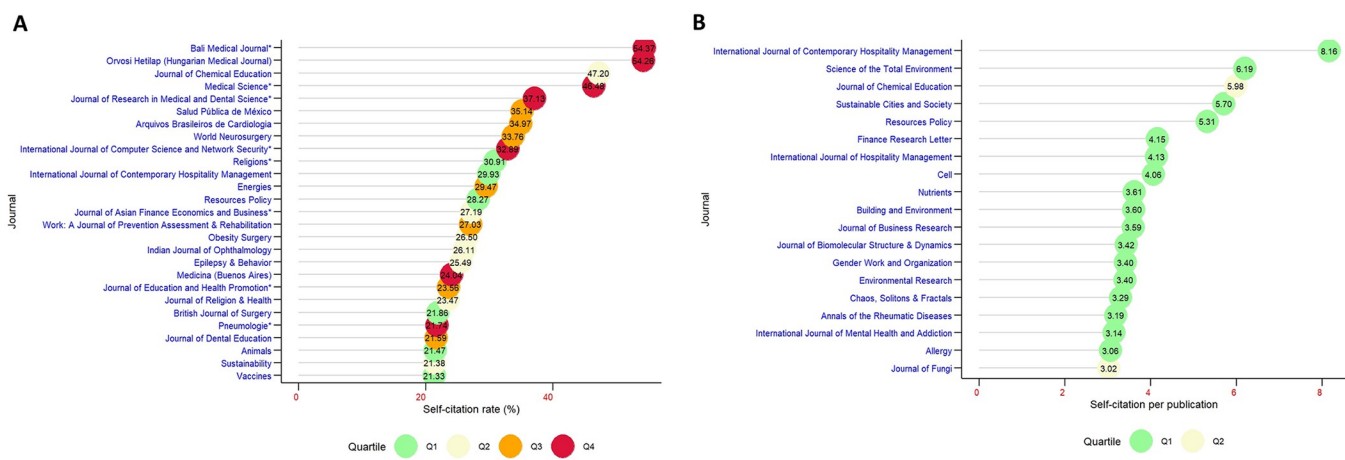

**Fig 3.** Lollipop charts of journals with the highest self-citation rates (A) and self-citation per publication (B) in COVID-19 research.

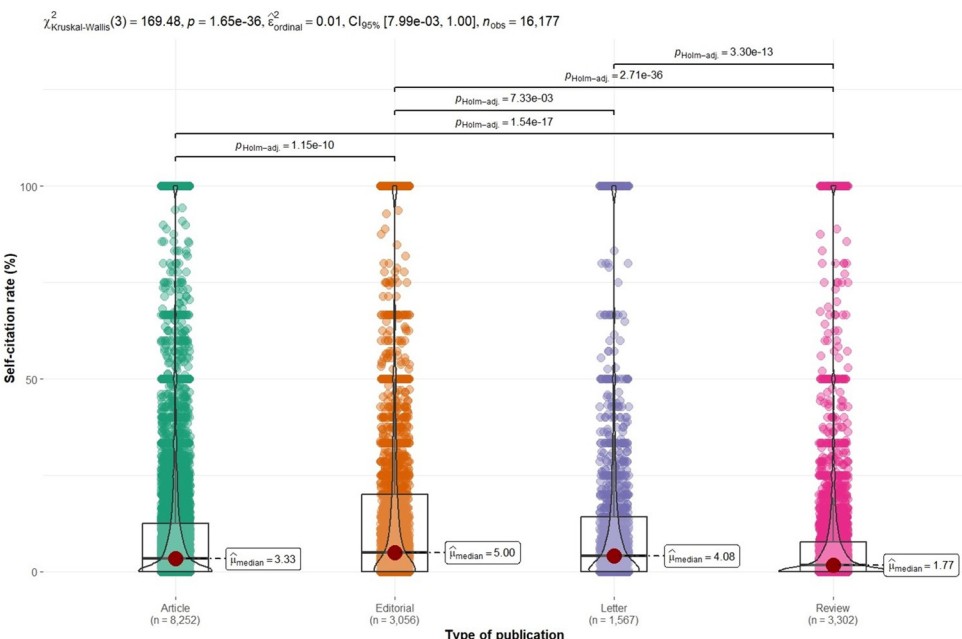

**Fig 4. Violin plots of journal self-citation rates in COVID-19 research according to the type of publication.**

statistical difference (p<0.001) when comparing journal SC per publication with those of COVID-19.

## Discussion

This exploratory study of journal SC in COVID-19 research indicated that 1,859 journals (13% of total coverage) had SCRs at or above 20%, meaning that more than one in five references are journal SCs and could be considered as journals with "high self-citation rates", the latter as previously suggested by Clarivate Analytics [42]. Also, the median SCR was 4.0% and the median JSC rate was 5.9%. The SCR represents what percentage of the total citations of a given journal are self-citations, while the JSC rate indicates the percentage change in a journal's IF when self-citations are not included compared to its baseline IF with self-citations. Those

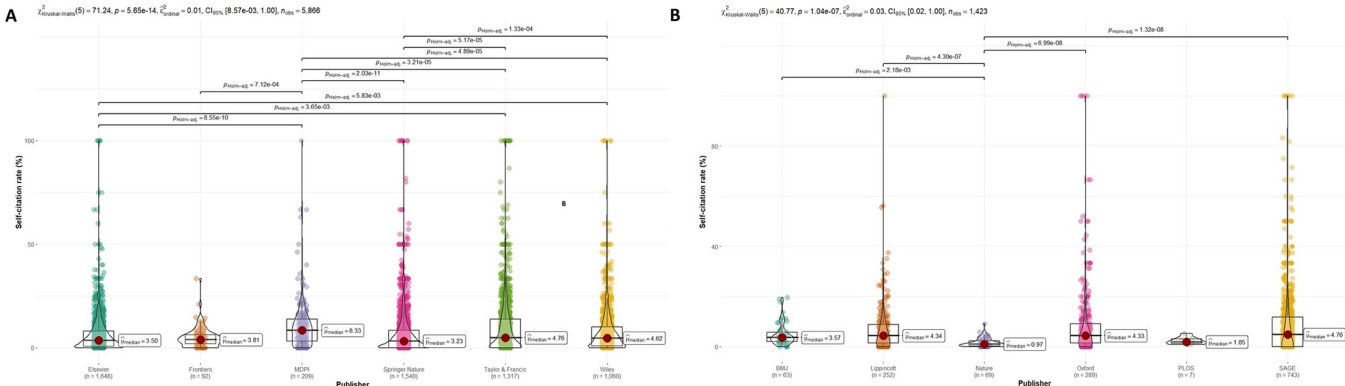

**Fig 5.** Violin plots of journal self-citation rates in COVID-19 research according to the most productive publishers: ranking 1st-6th (A) and ranking 7th-12th (B).

**Table 3. Comparison of journal self-citations in COVID-19 with other infectious diseases.**

| Timespan | Infectious Disease | Publications | Journals | Citations | Self-citation | Self-citation rate (%) | Self-citation/publication | p |
|---|---|---|---|---|---|---|---|---|
| 2020–2023 | COVID-19 | 377,343 | 14,690 | 6,167,073 | 307,080 | 4.98 | 0.81 | - |
| 2020–2023 | Tuberculosis | 23,379 | 3,225 | 102,381 | 6,731 | 6.57 | 0.29 | <0.001 |
| | HIV/AIDS | 21,846 | 3,137 | 102,093 | 7,009 | 6.87 | 0.32 | |
| 2016–2023 | Tuberculosis | 42,064 | 4,078 | 402,614 | 21,809 | 5.42 | 0.52 | <0.001 |
| | HIV/AIDS | 44,806 | 4,249 | 493,388 | 28,031 | 5.68 | 0.63 | |

values were lower than those reported in other areas such as dentistry, ophthalmology, dermatology (median SCR of 13.7, 11.7 and 10.5%, respectively), critical care medicine, anesthesiology, and general surgery (median JSC rate of 8.8, 8.4 and 6.5%, respectively) [23, 24, 26, 31, 34, 35]. Given that the present study included information from more than 14,000 journals, these discrepancies could be explained by the different sample sizes (< 100 journals) of these studies or their interest in analyzing very specific research areas.

Journals that belong to 2 or more JCR categories have higher SCR (7.7% vs. 4.7%) and JSC rates (7.4% vs. 4.4) than those that belong to a specific category, although this was not the case for fully open access journals. These particular findings could not be extrapolated to all COVID-19 journals, but only to the most productive journals. However, COVID-19 research embodied the urgency of contemporary global challenges, demanding dissemination avenues that match its gravity. On that note, mega-journals, characterized by their broad scope, interdisciplinary nature, and commitment to open access, serve as ideal platforms for the dissemination of such research [43]. Their broad publishing scope with acceptance rates ranging from 20% to 70%, and usually rapid review process make them attractive to the vast number of authors in COVID-19, although some of these journals also maintain relatively high SCRs [44]. It is curious to highlight the case of IJERPH, which has had meteoric growth since 2020 regarding the number of papers, accounting for around 15% of the papers in SCImago Journal & Country Rank *Public Health*, *Environmental and Occupational Health* category in 2021 [45]. However, since March of 2023 it was delisted from Clarivate and did not receive the 2022 IF due to the fact that some papers were declared outside the scope of the journal [46], making it impossible to calculate the JSC rate of the most productive journal in COVID-19 after four years of the start of the pandemic. For the same reason, its true impact in 2023 could not be fully calculated.

Journals with higher IF that have published in COVID-19 tend to exhibit higher SCs. This finding is consistent with results reported by Miyamoto in the field of plastic and reconstructive surgery [19]. However, several other studies have reported contrary findings, with some observing an inverse correlation in areas like pediatrics, dermatology, and ecological sciences [27, 33, 34], while others, such as in otolaryngology and emergency medicine, found no significant correlation [17, 21]. These discrepancies may be attributed to methodological differences, including variations in sources, the number of journals analyzed, and the time periods studied as depicted in Table 1. Although there were several other statistically significant correlation results between the journal SC and various indicators, one must carefully interpret these relationships since the correlation does not assure that the relationship between two variables is causal [47]. On the other hand, journals that belong to the lower quartiles (Q3 and Q4, based on a lower IF) are the ones that benefit the most from SC, and therefore the most vulnerable to changes in position when SC are withdrawn [48], but they are also the ones with the highest SCR [14, 15]. Fig 3 shows that the journals with the highest SCR belong to Q4 or Q3, however, there are also journals from higher quartiles. Also, when ordering the journals according to the number of SC per publication, only high-impact journals (Q1 or Q2) are observed. I

hypothesize that this difference is due to the denominator of the indicator, since in the SCR, despite a high number of SC, this is practically diluted in the Q1 or Q2 journals since their total citations are higher, while in the SC/publication only depends on the productivity of the journal, so they would be more comparable.

In the present study, editorials were the type of document that had the greatest number of SCs. Despite uncertainties about the ethics of allowing editors to publish in their own journals, this practice persists without defined regulations on the matter [49]. The IF of a journal for any specific year is calculated by dividing the total number of citations received by the articles published in the journal during the preceding 2 years (numerator), by the total number of articles published in the same 2 years (denominator) [50]. While all the published materials (original papers, reviews, editorials, letters to editor, news, book reviews, correspondence, etc.) are accepted in the numerator, only original papers and review articles are counted in denominator [50]. Therefore, having published several editorials with high percentages of SC could have inflated the 2022 IF since document types such as editorial materials and letters are not counted as 'citable items' in the denominator, but their citations are counted in the numerator of the IF calculation [14]. Although the present study cannot conclude that journal SC in COVID-19 caused by editorial materials have produced a dramatic change in the IF, it is undeniable that their SC levels are high. Previous research indicated that editorial materials from WoS did not seem to be extensively used to increase the IF by citing items published in the same journal [51]. Nonetheless, another study suggested that at least in other more specific fields, such as trauma and orthopedics, there is an increase in the IF score by 0.03 points, for every journal SC [10]. In the present study, the letters also had a large number of SCs. Manley demonstrated that the journal IF is reduced when letters are considered 'citable documents' [52]. Although his data is limited to 6 journals related to scholarly publication and metrics, his study showed that the smaller the number of citable items, the greater the impact on the IF calculation generated by even a single letter [52]. Letters are easier to accept in various journals and do not take much time to prepare as an original article even though they may sometimes report original data [53]. Indeed, there is a recent trend among several journals to prioritize using letters for disseminating studies that are less robust or have a smaller sample size and therefore require less extensive discussion [54]. For example, some high-impact journals like *JAMA* have the 'research letter' as a publication type that consists of focused reports of original research and can include any of the study types listed under original investigation. Similarly, *The Lancet* encourages the publication of some original results under the 'correspondence' category. Previous research showed that the contribution of citations from the so-called 'non-citable items' can increase the journal IF, especially in journals whose number of citable and non-citable items is unequal [55, 56]. Thus, the journal IF would decrease from the original if its calculation included all document types in the denominator or if citations of non-citable items were not counted in the numerator [14]. Although one can argue that the quality of an article's research is the main reason why a given article is cited or not, there is a specific Matthew effect associated with journals and this gives articles published there an added value above of its intrinsic quality [57, 58]. It is possible that in this case journal SCs are the initial impulse for journals that are not high impact to overcome the Matthew effect, and thus be able to attract citations to their papers independently of their own impact but rather due to the journal's status [14, 16]. However, this does not mean that high-impact journals do not have excessive SCs. Ultimately, like any other niche field, COVID-19 can have high SCs [24, 30, 34, 35].

Another interesting finding is the SCR at the publisher level, in which MDPI stands out significantly compared to the rest, although it had fewer journals (n = 209) with literature relevant to COVID-19 compared to other publishers with more than 1,000 journals. This may be

partially explained by the fact that MDPI has several journals that are among the most productive in COVID-19 (Table 2), and since the relationship between the number of publications and SCs is proportional, it is expected to obtain these results. However, this publisher has already been questioned before, not only for high SCR in its journals, but also for an increasingly high rate of citations from other MDPI-journals, thus creating an intra-publisher citation network [59, 60], and also for its countless articles in special collections compared to other publishers [46]. The study by You et al. suggested that questionable publishers inflate the impact of their citations by including publisher-level SCs, which poses challenges for detection through traditional journal metrics [61]. This practice is also known as citation stacking [62], and it has become even more complex when intra- and inter-disciplinary publisher SCs come into play [63]. There is no better example of interdisciplinarity in recent years than COVID-19, mobilizing research in various areas of knowledge in a short time, which is associated with the impact of research based on its citations, and in turn with SCs [62, 64]. Exacerbated by a constant race against the clock to publish quickly and massively, this is also reflected when comparing COVID-19 with other long-standing diseases. In the present study, the level of journal SCRs was lower than the rest due to the exorbitant number of citations in the denominator, although at the same time, the number of SCs per publication stands out, which was significantly higher in both periods. One should approach with caution when comparing different infectious diseases, taking into account the rapid, pandemic spread of COVID-19 and the varying impact of these distinct disease entities.

Through a survey of more than 110,000 scholars from various disciplines and using a journal-based dataset from Scopus, a study concluded that higher-ranked journal editors seem to tend to force scholars to add citations that are not pertinent to their work and that commercial publishers are more likely to own journals that have coerced [65]. Although these results are definitely not conclusive, concerns are raised about how "relevant" the SC of some journals were. Journal SC are a natural part of the publication process, and though manipulation of citations is unethical [66], there is no clear limit as to how much is acceptable. This study does not intend to conclude that all SCs from journals and publishers correspond to coercive SCs, but rather its findings shed light on an even bigger question: how to detect coercive self-citations and how to stop them? Although some authors have developed sophisticated algorithms to detect journals or publishers SC [67, 68], it is still difficult to accurately distinguish between legitimate citations and those intended to artificially inflate impact metrics.

To address the issue of problematic self-citation practices, authors, reviewers, editors, and publishers can implement several specific measures:

1. Exclude journal SCs from the IF calculation since this would eliminate the motivation to coerce [69].

2. Adopt measures such as a transparent peer review, so that the reviewers' reports, authors' responses, and editors' decision letters are all published alongside an article [70]. Transparency can discourage unethical practices and allow the community to monitor and report any irregularities.

3. Avoid the inappropriate use of 'non-citable items' for the calculation of IF, since all material published in a journal is subject to being cited, in this way all document types must be included in the denominator of the IF calculation [14].

4. Report the classic journal impact metrics together with the level of journal SCs, similar to what was proposed for the authors' h-index [71].

5. Foster a community culture that values integrity over impact metrics along with publishers establishing ethical guidelines and impose sanctions to those who engage in such practices [65]. For instance, by imposing a ban on authors, removing editors from editorial roles, delisting reviewers from journals' list of reviewers, and blacklisting journals from citation indexes [72].

6. Rely on peer reviewers who are crucial in maintaining the scientific quality and integrity of published papers. However, due to the high volume of submissions and their busy schedules, they may not always verify every referenced source [72, 73]. Editors, who approve reviewers' recommendations, are ultimately responsible for any inappropriate suggestions [73].

According to a document released by COPE (the Committee on Publication Ethics) in July 2019, citation manipulation refers to excessive citation of articles from the journal in which the author is publishing an article or another journal ('citation stacking') as a means solely of increasing the number of citations of the journal in question [74]. There are many instances where citing researchers' own work or that of colleagues are legitimate. This action becomes unethical only when the cited sources are not materially relevant to the manuscript's scholarly content [72, 75]. Additionally, there are distorted publishing incentives that can lead scientists to publish for the wrong reasons, often chasing a journal IF score for a paper rather than producing well-structured and valuable research for their peers, regardless of where it's published [73]. Unfortunately, metrics like the IF are used for formal evaluations and promotions of scientists in academia, particularly in developing nations, and are thus open to wide abuse by authors, editors and journals [50, 73]. Although, IF is often erroneously used to judge the quality of a paper, it was meant to exclusively judge the "popularity" of a journal [73]. Weale et al. suggested using the level of non-citation of articles within a journal as an alternative measure, as the ultimate goal of all research is to be cited in other authors' work. This metric also provides a more comparable measure across different disciplines [76]. Finally, a new phenomenon is emerging where articles are cited in exchange for monetary compensation, aimed at manipulating reference lists to inflate a journal's IF [72, 77].

This study has several limitations. First, preprints were not considered, even though they were a means of important and rapid dissemination of valuable information amid the threat of the pandemic, especially in the early phase [78]. However, not considering them allowed me to avoid duplicates and only select information that has already been peer-reviewed and published in a scientific journal. Second, the study did not cover journals not listed in the JCR. The JCR contains more than 21,000 journals and although it does not have many journals compared to other databases, its selectivity feature allows it to include only journals that met high-research standards and passed a strict process to continue their indexing in the JCR, and avoid being delisted each year [37, 39, 79]. Thus, the findings of the present study do not apply to other journals that may have published COVID-19 papers in other databases. Overall, the results of journal SC in COVID-19 research have a high degree of generalizability to all WoSCC journals due to the key characteristic of having used a complete sampling frame of journals that provides stronger and more reliable conclusions. Nonetheless, it is pertinent to recognize that including only the top twelve publishers based on their number of publications raises concerns that would limit the generalizability of publisher SC in COVID-19 research and could potentially result in misleading conclusions. Finally, there is a citation and indexing time lag of publications in the WoSCC that could have affected the findings. Future studies should aim to expand these datasets, particularly by completing the dataset for publishers in COVID-19 research, as well as to characterize the journal SC trends according to the various research fields involved in COVID-19 research.

## Conclusions

In summary, this study has shown that the median SCR was 4% and that in 13% of the journals, one in five references was a journal self-citation. The number of publications, citations, citations per publication correlated with journal self-citations. The editorial materials contributed more to journal self-citations, suggesting that in COVID-19 research, self-citing non-citable items could potentially contribute to inflate journal impact factors. The MDPI publisher, despite having less journals in the topic, had higher self-citation rates compared to other publishers with more journals. The number of SC per publication of COVID-19 research was higher than those for tuberculosis and HIV/AIDS.

## Supporting information

**S1 File. Search strategy.**
(DOCX)

**S2 File. Journals, type of publication and publisher datasets based in COVID-19 research.**
(XLSX)

**S3 File. Tuberculosis and HIV/AIDS research.**
(XLSX)

## Author Contributions

**Conceptualization:** Alvaro Quincho-Lopez.

**Data curation:** Alvaro Quincho-Lopez.

**Formal analysis:** Alvaro Quincho-Lopez.

**Methodology:** Alvaro Quincho-Lopez.

**Software:** Alvaro Quincho-Lopez.

**Supervision:** Alvaro Quincho-Lopez.

**Visualization:** Alvaro Quincho-Lopez.

**Writing – original draft:** Alvaro Quincho-Lopez.

**Writing – review & editing:** Alvaro Quincho-Lopez.

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
