## [Decision Letter · Decision Letter 0]

5 Jun 2024

PONE-D-24-11143Comparison of self-citation rates among journals and publishers on COVID-19 researchPLOS ONE

Dear Dr. Quincho-Lopez,

Thank you for submitting your manuscript to PLOS ONE. After careful consideration, we feel that it has merit but does not fully meet PLOS ONE’s publication criteria as it currently stands. Therefore, we invite you to submit a revised version of the manuscript that addresses the points raised during the review process.

We look forward to receiving your revised manuscript.

Kind regards,

Tauseef Ahmad, PhD

Academic Editor

PLOS ONE

Journal Requirements:

Reviewers' comments:

Reviewer's Responses to Questions

**Comments to the Author**

1. Is the manuscript technically sound, and do the data support the conclusions?

Reviewer #1: Yes

Reviewer #2: Yes

Reviewer #3: Yes

Reviewer #4: Yes

Reviewer #5: Yes

2. Has the statistical analysis been performed appropriately and rigorously? 

Reviewer #1: Yes

Reviewer #2: N/A

Reviewer #3: Yes

Reviewer #4: Yes

Reviewer #5: Yes

3. Have the authors made all data underlying the findings in their manuscript fully available?

Reviewer #1: Yes

Reviewer #2: Yes

Reviewer #3: Yes

Reviewer #4: Yes

Reviewer #5: Yes

4. Is the manuscript presented in an intelligible fashion and written in standard English?

Reviewer #1: Yes

Reviewer #2: Yes

Reviewer #3: Yes

Reviewer #4: Yes

Reviewer #5: Yes

5. Review Comments to the Author

**Reviewer #1:** The article has two challenges.

1. The statement of the problem is not clear.

2. Conclusions and analyzes are not enough and practical suggestions should be presented.

3. In this article, literature review and its summary is not observed and it is better to add a part to it.

**Reviewer #2: **This study provides an important analysis of journal self-citation rates on COVID-19 research, which is a timely and relevant topic given the rapid publication of COVID-19 studies and concerns about potential manipulation of citation metrics. The authors present a comprehensive dataset and some interesting findings, such as the high self-citation rates of certain journals and publishers. However, there are several significant weaknesses that need to be addressed through major revisions.

Expand the literature review to provide more context on journal self-citation practices, including comparisons to self-citation rates in other research areas. This will help readers better interpret the results.

Expand the discussion section to thoroughly explore potential reasons for the observed self-citation patterns, drawing on existing literature and theories about citation manipulation. Discuss the role of editorial practices, publisher incentives, and other factors that may contribute to high self-citation rates.

Add a section with concrete recommendations for addressing problematic self-citation practices, such as changes to editorial policies, peer review processes, citation metrics, or industry-wide initiatives to promote integrity.

Expand the limitations section to thoroughly discuss the study's weaknesses and acknowledge areas where the findings may be incomplete or biased. Suggest directions for future research that could build on this work and address the identified limitations.

Overall, this is an important study that sheds light on a concerning issue in scholarly publishing. With major revisions to provide more context, explore potential causes in-depth, offer recommendations, and thoroughly discuss limitations, the paper could make a valuable contribution to the literature on citation practices and journal integrity.

**Reviewer #3: **Manuscript is well written and investigates interesting topic. However, some clarifications should be made.

I think title should be “journal and publisher self-citation” not “self-citation among” because you did not investigate author self-citation. I think author should be very specific about this. For instance, in the conclusions rows 54, 322 and 323, there is room for misinterpretation.

Row 98, what ”different variants”?

I don’t understand difference between SCR and JCR described in the methods. I would recommend you also repeat what they are and how they are interpreted in plain language in the beginning of discussion.

I think it would be important to know what are “research areas”?

So, self-citations are not only on citations to COVID-related articles but self-citations in COVID-related articles? If this is true, I think it is good to state already in the methods, and be specific about that in all instances.

Spearman correlation measures linear correlation. Is that only relationship you are interested of? Did you check scatter plots? Or consider performing some adjusted analyses. E.g. publication time is likely influential issue here.

Table 2: open abbreviations.

Row 227, check grammar or clarify.

Row 243, is 20% high acceptance rate?

Row 252, so low quartiles are Q4 and Q3?

Row 314, any more detail information about coverage? Give an evaluation about the generalizability of your findings to all COVID-related research.

**Reviewer #4: **The article "Comparison of self-citation rates among journals and publishers on COVID-19 research" needs several points of revision for better impact.

1. The statement "Some journals display exorbitant self-citation patterns" in the abstract's conclusion might be too generalized based on the specific data sample. It might lead to an overgeneralization fallacy if the study's scope needs to be broadened to represent all journals fairly.

2. The introduction mentions that "the scientific community responded quickly by generating new publications at an unprecedented level" and implies a direct correlation to citation rates without considering other factors. There is an assumption that rapid publication directly leads to higher self-citation rates without investigating other potential causes.

3. The study focuses on data from the Web of Science Core Collection and specific filters. This might exclude relevant data from other databases or journals that do not index with WoSCC, introducing a potential sampling bias. Reasons for excluding Scopus or Pubmed could be mentioned.

4. The results mention a "positive and statistically significant correlation of self-citations with the other indicators," but this does not necessarily imply causation. The manuscript should be cautious in interpreting these correlations as causal relationships.

5. The study finds many results statistically significant (p<0.001), but it does not discuss the practical significance or the effect size of these findings. Statistical significance does not always imply that the findings are practically meaningful.

6. The discussion suggests that "self-citing non-citable items could potentially contribute to inflate journal impact factors during the pandemic." This is true for any niche topic, and self-citation is the initial push for any glass ceiling or overcoming the "Mathew effect." It has been previously detailed in "Deora H, Kraus KL, Couldwell WT, Garg K. Self-Citation Rates Among Neurosurgery Journals and Authors: Unethical or Misunderstood? World Neurosurg. 2023 Oct;178:e307-e314. doi: 10.1016/j.wneu.2023.07.052. Epub 2023 Jul 18. PMID: 37473867."

7. The study focuses on the top twelve publishers and specific high-impact journals. This selection might lead to cherry-picking, where only data supporting the hypothesis are presented, while contrary evidence is ignored.

9. While the study is rich in statistical analysis, it lacks qualitative insights that could provide context to the quantitative data. Understanding the reasons behind self-citations requires more than just numerical data; it needs context from editorial practices and author motivations.

While self-citation is an issue, Covid was a niche topic and may have higher self-citations, so the other side of the coin needs to be mentioned.

**Reviewer #5: **Dear Authors,

Greetings,

I have reviewed your manuscript titled "Comparison of self-citation rates among journals and publishers on COVID-19 research." Your work addresses a significant topic in current bibliometric analysis. Please find my detailed feedback attached.

Best regards.

1. Technical Soundness and Data Support for Conclusions

The manuscript describes a thorough analysis of self-citation rates among journals and publishers on COVID-19 research, using data extracted from the Web of Science Core Collection and analyzed with InCites. The methodology appears robust, employing appropriate statistical comparisons and controls, such as the exclusion of non-relevant document types and the use of various bibliometric indicators.

The data supports the conclusions adequately. The results section provides detailed statistics and comparisons that align with the conclusions drawn, such as the identification of journals with high self-citation rates and the correlation between self-citations and other bibliometric indicators.

2. Appropriateness and Rigor of Statistical Analysis

The statistical analysis in the manuscript is rigorous and appropriate. The authors used a range of statistical methods, including the Mann–Whitney U-test, Kruskal-Wallis test, and Spearman's rank correlation coefficient, which are suitable for the non-parametric data and the comparisons made. The use of violin plots and lollipop charts enhances the clarity of the statistical findings.

Suggestions:

- Ensure all statistical tests are clearly explained in the methods section.

- Confirm that all assumptions of the statistical tests used are met, and report any deviations.

3. Data Availability

The authors state that all data underlying the findings are fully available without restriction, in compliance with PLOS ONE's data policy. The supporting information files (S1 File, S2 File, and S3 File) are provided, which include the search strategy and datasets used for analysis.

Suggestions:

- Ensure that the data availability statement is clear and specific about where the data can be accessed, including URLs or DOIs for datasets in public repositories.

4. Manuscript Presentation and Language

The manuscript is presented in an intelligible fashion and written in standard English. However, there are some typographical and grammatical errors that need correction to improve clarity and readability.

Suggestions:

- Proofread the manuscript to correct typographical and grammatical errors.

- Consider simplifying complex sentences to improve readability for a broader audience.

5. Review Comments to the Author

Strengths:

- The manuscript addresses a relevant and timely topic, providing insights into the self-citation practices of journals and publishers during the COVID-19 pandemic.

- The use of multiple bibliometric indicators and robust statistical analysis methods strengthens the validity of the findings.

Weaknesses:

- While the methodology is generally sound, it relies heavily on data from the Web of Science Core Collection and InCites, which may introduce biases related to the coverage and indexing practices of these databases.

- The discussion section could be expanded to include a more in-depth analysis of the implications of high self-citation rates, particularly in the context of research integrity and journal impact metrics.

Recommendations:

To summarize, these are my recommendations:

1. Expand Discussion: Provide a more detailed discussion of the potential consequences of high self-citation rates, including ethical considerations and the impact on journal metrics.

2. Clarify Data Availability: Ensure the data availability statement clearly specifies how and where the data can be accessed, including any relevant URLs or DOIs.

3. Improve Readability: Address typographical and grammatical errors throughout the manuscript to enhance readability.

6. PLOS authors have the option to publish the peer review history of their article (what does this mean?). If published, this will include your full peer review and any attached files.

Reviewer #1: No

Reviewer #2: **Yes: **Dr. Mueen Ahmed KK

Reviewer #3: No

Reviewer #4: No

Reviewer #5: No

---

## [Author Response · Author response to Decision Letter 0]

14 Aug 2024

August 13, 2024

I am deeply grateful to the Editor and all reviewers for such valuable comments. In the following lines you may find the responses to each comment. 

Reviewer #1: The article has two challenges.

1.- The statement of the problem is not clear.

Thanks for the comment. The last paragraph of the introduction was paraphrased. Please, see lines 115 to 122.

2.- Conclusions and analyzes are not enough and practical suggestions should be presented.

Thanks for the suggestion. Six concrete suggestions were added in the discussion section. Please, see lines 423 to 449.

3.- In this article, literature review and its summary is not observed and it is better to add a part to it. 

Thanks for the suggestion. A brief key literature review was added from lines 92 to 113. Also, a table (Table 1) comparing different aspects of those studies (source, period, number of journals and how data was collected, main findings and others) was added.

Reviewer #2: This study provides an important analysis of journal self-citation rates on COVID-19 research, which is a timely and relevant topic given the rapid publication of COVID-19 studies and concerns about potential manipulation of citation metrics. The authors present a comprehensive dataset and some interesting findings, such as the high self-citation rates of certain journals and publishers. However, there are several significant weaknesses that need to be addressed through major revisions.

1.- Expand the literature review to provide more context on journal self-citation practices, including comparisons to self-citation rates in other research areas. This will help readers better interpret the results.

Thanks for the suggestion. A brief key literature review including comparisons of self-citations in different research fields was added from lines 92 to 113. Also, a table (Table 1) comparing different aspects of those studies (source, period, number of journals and how data was collected, main findings and others) was added. 

2.- Expand the discussion section to thoroughly explore potential reasons for the observed self-citation patterns, drawing on existing literature and theories about citation manipulation. Discuss the role of editorial practices, publisher incentives, and other factors that may contribute to high self-citation rates.

Thanks for the comment. Some lines (357 to 380) were added with respect to the use of ‘non-citable’ items, for instance, editorials and letters, and how they can be used to influence the journal self-citation rates and therefore inflate the journal metrics. A new paragraph regarding the editorial practices like the erroneous use of journal IF and incentives in academia was added in the discussion section. Please, see lines 450 to 467. However, it should be clear that this study cannot conclude that all journal self-citations are due to citation manipulation.

3.- Add a section with concrete recommendations for addressing problematic self-citation practices, such as changes to editorial policies, peer review processes, citation metrics, or industry-wide initiatives to promote integrity.

Thanks for the suggestion. Six concrete suggestions were added in the discussion section. Please, see lines 423 to 449.

4.- Expand the limitations section to thoroughly discuss the study's weaknesses and acknowledge areas where the findings may be incomplete or biased. Suggest directions for future research that could build on this work and address the identified limitations.

Thanks for the comment. The limitations section was expanded and future research directions were added accordingly. For instance, I acknowledge the fact that focusing on the top publishers could lead to misleading conclusions. Please, see lines 473 to 487. 

Overall, this is an important study that sheds light on a concerning issue in scholarly publishing. With major revisions to provide more context, explore potential causes in-depth, offer recommendations, and thoroughly discuss limitations, the paper could make a valuable contribution to the literature on citation practices and journal integrity.

Reviewer #3: Manuscript is well written and investigates interesting topic. However, some clarifications should be made.

1.- I think title should be “journal and publisher self-citation” not “self-citation among” because you did not investigate author self-citation. I think author should be very specific about this. For instance, in the conclusions rows 54, 322 and 323, there is room for misinterpretation.

Thanks for the comment. I agree. The new title is as follows: “Comparison of journal and top publisher self-citation rates in COVID-19 research”. Some clarifications were in made in rows 58, 491 and 492 accordingly. 

2.- Row 98, what “different variants”?

By that I meant synonyms/search terms of the word ‘COVID-19’. In order to clarify that, the word ‘variants’ was replaced by ‘search terms’ Please, see row 132.

3.- I don’t understand difference between SCR and JCR described in the methods. I would recommend you also repeat what they are and how they are interpreted in plain language in the beginning of discussion.

Thanks for the suggestion. I think by JCR you meant JSC rate because the JCR stands for the Journal Citation Report, which is a resource developed by Clarivate Analytics that provides metrics and data on the impact of journals from the Web of Science Core Collection. 

The self-citation rate (SCR) is just the percentage of journal self-citation. For example, 25 journal self-citations out of 100 citations would be a SCR of 25%.

The journal self-citation (JSC) rate is based on how the impact factor of a journal changes with and without journal self-citations. The term was coined by Sanfilippo in some articles, like this: https://www.ncbi.nlm.nih.gov/pmc/articles/PMC7836441/. The formula to calculate the JSC rate is as follows: JSC rate = (IF – IF without self-citations) / IF.

The methodology section was modified accordingly. Please, see lines 165 to 166. I also followed your recommendation, and in the discussion section, I have repeated what they are and how they can be interpreted. Please, see lines 292 to 294. 

4.- I think it would be important to know what are “research areas”?

By ‘research areas’ I meant the research category assigned by the Journal Citation Reports. For instance, the journal Vaccines according the Web of Science Core Collection has two JCR categories, each one could have different quartiles or the same quartile.

The following changes were made:

Row 167-168: “the number of categories assigned by the JCR and…”.

Row 220 and 301: “Journals that belong to 2 or more JCR categories…”.

5.- So, self-citations are not only on citations to COVID-related articles but self-citations in COVID-related articles? If this is true, I think it is good to state already in the methods, and be specific about that in all instances.

Thank you for your comment. I appreciate your suggestion to clarify the definition of self-citations used in this study. In response, I ensured that my definition of self-citations aligns with the second definition you highlighted: self-citations in COVID-related articles. This means that I am focusing on instances where articles about COVID-19 published in a journal cite any previous work from the same journal, regardless of whether those cited articles are about COVID-19 or other topics. Somes lines were added in the methods section to state this definition (row 153 to 155) Additionally, I revised the manuscript to ensure that this specific definition is consistently applied accordingly.

6.- Spearman correlation measures linear correlation. Is that only relationship you are interested of? Did you check scatter plots? Or consider performing some adjusted analyses. E.g. publication time is likely influential issue here.

Thanks for the suggestion. However, Spearman correlation is used to detect monotonic relationships but not linear relationships. In a monotonic relationship the variables tend to move in the same direction, but not necessarily at a constant rate like linear relationships where changes in one variable influence proportional changes in another variable. Performing adjusted analyses is beyond the scope of the present study. The idea of controlling confounding factors in adjusted analyzes sounds interesting, but that data is not easily accessible. Other factors to consider would be the specialization of the journal, the number of issues per year, whether the journal allows preprints publication prior of the full version, and others. For example, the only alternative to incorporating publication times would be to manually search for the publication time in each article and average them to obtain a single value for each journal, which would be very laborious for more than 14,000 journals. My response is based on this article: https://pubmed.ncbi.nlm.nih.gov/29481436/

7.- Table 2: open abbreviations.

Those changes were made in Table 3 (former Table 2) accordingly.

8.- Row 227, check grammar or clarify.

Thanks for noting that grammar mistake. Now, in row 289 the text is as follows: “This exploratory study of journal SC on COVID-19 research indicated that approximately more than one in five references are journal SCs and that thirteen out of every hundred…”.

9.- Row 243, is 20% high acceptance rate?

Thanks for noticing that. I am paraphrasing that sentence so it would not imply that 20% is a high acceptance rate when reading. Now, in row 308, the text is as follows: “Their broad publishing scope with acceptance rates ranging from 20% to 70%, …”. 

10.- Row 252, so low quartiles are Q4 and Q3?

Yes, if quartiles are divided in two parts, there are journals that belong to lower quartiles (Q3 and Q4) while others belong to higher quartiles (Q1 and Q2). Please, see line 337.

11.- Row 314, any more detail information about coverage? Give an evaluation about the generalizability of your findings to all COVID-related research.

Thanks for the suggestion. I agree. Now, the discussion section provides details on the coverage, in addition to the generalizability of the results to journals that have published on COVID-19, and recognizes the limitations regarding the publishers. Please, see lines 463 to 476. 

Reviewer #4: The article "Comparison of self-citation rates among journals and publishers on COVID-19 research" needs several points of revision for better impact.

1.- The statement "Some journals display exorbitant self-citation patterns" in the abstract's conclusion might be too generalized based on the specific data sample. It might lead to an overgeneralization fallacy if the study's scope needs to be broadened to represent all journals fairly.

Thanks for the suggestion. I agree, then the abstract (row 59 to 60) was modified as follows: “Some journals from the Web of Science Core Collection displayed exorbitant journal self-citation patterns during the period 2020-2023”. 

2.- The introduction mentions that "the scientific community responded quickly by generating new publications at an unprecedented level" and implies a direct correlation to citation rates without considering other factors. There is an assumption that rapid publication directly leads to higher self-citation rates without investigating other potential causes.

Thanks for your comment. However, in that section of the introduction I was just describing what the reference (Brandt et al.) claims (https://www.ncbi.nlm.nih.gov/pmc/articles/PMC9328554/). The literal text of that paper is as follows: “Amidst all of this, there has been an explosion of peer-reviewed literature about Covid-19 as researchers work to uncover details such as structure, infectivity, spread, effects, prevention, and treatment of this novel virus.”. Actually, that study minimized the influence of other factors like article number variation of each journal and field of research when performing a binomial regression. Despite this, the study concludes that Covid-19 papers have still experienced more than 80% increase in citations relative to non-Covid-19 papers. 

No assumption was made about the relationship between rapid publication and self-citation, even no mention of the word ‘self-citation’ was made in that paragraph of the introduction. However, in order to avoid any misinterpretation of that paragraph, the following text (row 67) was added as follows: “Unsurprisingly, those publications have been cited by multiple other papers at incredibly high rates while the pandemic continues to be in the spotlight.”. 

3.- The study focuses on data from the Web of Science Core Collection and specific filters. This might exclude relevant data from other databases or journals that do not index with WoSCC, introducing a potential sampling bias. Reasons for excluding Scopus or Pubmed could be mentioned.

Thanks for the suggestion. I agree. In the methods section, please see lines 127 to 131.

4.- The results mention a "positive and statistically significant correlation of self-citations with the other indicators," but this does not necessarily imply causation. The manuscript should be cautious in interpreting these correlations as causal relationships.

Thanks for the suggestion. In order to clarity to the readers any misinterpretation of the correlation results, I added some lines (row 334 to 336) in the discussion section: “Although there were several statistically significant correlation results between the journal SC and various indicators, one must carefully interpret these relationships since the correlation does not assure that the relationship between two variables is causal.”.

5.- The study finds many results statistically significant (p<0.001), but it does not discuss the practical significance or the effect size of these findings. Statistical significance does not always imply that the findings are practically meaningful.

Thanks for the comment. Now, in the discussion section I have added some lines regarding the correlation results (rows 327 to 334) and the role of ‘non-citable’ items (letters and editorials) and how they can influence on the calculation of the journal IF (359 to 384). In the original version, it was already discussed the publisher SC results and the comparison among infectious diseases and COVID-19.

6.- The discussion suggests that "self-citing non-citable items could potentially contribute to inflate journal impact factors during the pandemic." This is true for any niche topic, and self-citation is the initial push for any glass ceiling or overcoming the "Mathew effect." It has been previously detailed in "Deora H, Kraus KL, Couldwell WT, Garg K. Self-Citation Rates Among Neurosurgery Journals and Authors: Unethical or Misunderstood? World Neurosurg. 2023 Oct;178:e307-e314. doi: 10.1016/j.wneu.2023.07.052. Epub 2023 Jul 18. PMID: 37473867."

Thanks for your comment. That is a good point. Some lines regarding the Matthew effect were added. Please, see lines 375 to 384.

7.- The study focuses on the top twelve publishers and specific high-impact journals. This selection might lead to cherry-picking, where only data supporting the hypothesis are presented, while contrary evidence is ignored.

Thanks for the suggestion. However, I don’t think that data of more than 14,000 journals is regarded as just specific high-impact journals. All documents found with the search query were extracted from WoSCC. Table 2 shows only the main journals but supplementary material 2 shows data from all journals included in the present study. However, what I do agree with is the issue of the publishers, since the first twelve were chosen based on their productivity, ignoring the rest. For this reason, the following lines were added to the discussion (row 483 to 486): "Nonetheless, it is pertinent to recognize that including only the top twelve publishers based on their number of publications raises concerns that would limit the generalizability of publisher SC in COVID-19 research and could potentially result in misleading conclusions".

8.-. While the study is rich in statistical analysis, it lacks qualitative insights that could provide context to the quantitative data. Understanding the re

---

## [Decision Letter · Decision Letter 1]

22 Oct 2024

PONE-D-24-11143R1Comparison of journal and top publisher self-citation rates in COVID-19 researchPLOS ONE

Dear Dr. Quincho-Lopez,

Thank you for submitting your manuscript to PLOS ONE. After careful consideration, we feel that it has merit but does not fully meet PLOS ONE’s publication criteria as it currently stands. Therefore, we invite you to submit a revised version of the manuscript that addresses the points raised during the review process.

We look forward to receiving your revised manuscript.

Kind regards,

Tauseef Ahmad, PhD

Academic Editor

PLOS ONE

Journal Requirements:

**Additional Editor Comments:**

Dear author,

I will be happy to have made my decision to accept your paper titled "Comparison of journal and top publisher self-citation rates in COVID-19 research" for publication in PLoS One.

Before that, the author requested to clarify some of the information to avoid readers confusion and misleading. 

1) Use the full form of COVID-19 for the first time 

2) Explain a bit about the calculation of the Impact Factor, as you mentioned in the paper that the Editorial materials have the highest rate of citations, as the Editorial materials or correspondence citations are included in the Impact Factor calculation?

3) Add a flow chart of the data retrieval, extraction process and analysis.

We are looking forward to your resubmission!

Best regards

Reviewers' comments:

Reviewer's Responses to Questions

**Comments to the Author**

1. If the authors have adequately addressed your comments raised in a previous round of review and you feel that this manuscript is now acceptable for publication, you may indicate that here to bypass the “Comments to the Author” section, enter your conflict of interest statement in the “Confidential to Editor” section, and submit your "Accept" recommendation.

Reviewer #3: All comments have been addressed

Reviewer #5: All comments have been addressed

2. Is the manuscript technically sound, and do the data support the conclusions?

Reviewer #3: Yes

Reviewer #5: Yes

3. Has the statistical analysis been performed appropriately and rigorously? 

Reviewer #3: Yes

Reviewer #5: Yes

4. Have the authors made all data underlying the findings in their manuscript fully available?

Reviewer #3: Yes

Reviewer #5: Yes

5. Is the manuscript presented in an intelligible fashion and written in standard English?

Reviewer #3: Yes

Reviewer #5: Yes

6. Review Comments to the Author

Reviewer #3: The first sentence of discussion is misleading and is not in align with the abstract and conclusion:

"one in five references are journal SCs" was found in 13% of journals with the highest SC rate.

What if just using sentences from the abstract at the beginning? "The median self-citation rate was 4.0% (IQR 0–11.7%), and the median journal self-citation rate was 5.9% (IQR 0-12.5%). 1,859 journals (13% of total coverage) had self-citation rates at or above 20%, meaning that more than one in five references are journal self-citations."

Reviewer #5: Dear Authors,

Thank you for addressing the feedback provided. I have reviewed your revisions and find that you have adequately responded to the concerns raised, including expanding the discussion section, clarifying the data availability, and improving the manuscript's readability. I have no further comments or suggestions at this time.

Best regards,

7. PLOS authors have the option to publish the peer review history of their article (what does this mean?). If published, this will include your full peer review and any attached files.

Reviewer #3: No

Reviewer #5: No

---

## [Author Response · Author response to Decision Letter 1]

27 Oct 2024

October 27, 2024

I am deeply grateful to the Editor and the reviewers for the consideration of this manuscript. In the following lines you may find the responses to each comment. 

Editor:

Dear author,

I will be happy to have made my decision to accept your paper titled "Comparison of journal and top publisher self-citation rates in COVID-19 research" for publication in PLoS One. Before that, the author requested to clarify some of the information to avoid readers confusion and misleading. 

1) Use the full form of COVID-19 for the first time.

Thanks for the comment. I agree. Please, check line 59. 

2) Explain a bit about the calculation of the Impact Factor, as you mentioned in the paper that the Editorial materials have the highest rate of citations, as the Editorial materials or correspondence citations are included in the Impact Factor calculation?

Thanks for the comment. That explanation is now provided in lines 326-331.

3) Add a flow chart of the data retrieval, extraction process and analysis.

Thanks for the comment. I agree. I added a new Figure 1 as the flow chart (line 175), and rename the rest accordingly in the text. 

Reviewer #1:

The first sentence of discussion is misleading and is not in align with the abstract and conclusion:

"one in five references are journal SCs" was found in 13% of journals with the highest SC rate.

What if just using sentences from the abstract at the beginning? "The median self-citation rate was 4.0% (IQR 0–11.7%), and the median journal self-citation rate was 5.9% (IQR 0-12.5%). 1,859 journals (13% of total coverage) had self-citation rates at or above 20%, meaning that more than one in five references are journal self-citations."

Thanks for the comment. I agree. Please, check lines 270-272.

Reviewer #5: 

Dear Authors,

Thank you for addressing the feedback provided. I have reviewed your revisions and find that you have adequately responded to the concerns raised, including expanding the discussion section, clarifying the data availability, and improving the manuscript's readability. I have no further comments or suggestions at this time.

Thank you for the feedback provided.

---

## [Editor Report · Decision Letter 2]

20 Nov 2024

Comparison of journal and top publisher self-citation rates in COVID-19 research

PONE-D-24-11143R2

Dear Dr. Quincho-Lopez,

We’re pleased to inform you that your manuscript has been judged scientifically suitable for publication and will be formally accepted for publication once it meets all outstanding technical requirements.

Kind regards,

Tauseef Ahmad, PhD

Academic Editor

PLOS ONE
---

## [Editor Report · Acceptance letter]

25 Nov 2024

PONE-D-24-11143R2 

PLOS ONE

Dear Dr. Quincho-Lopez, 

I'm pleased to inform you that your manuscript has been deemed suitable for publication in PLOS ONE. Congratulations! Your manuscript is now being handed over to our production team.

Kind regards, 

on behalf of

Dr. Tauseef Ahmad 

Academic Editor

PLOS ONE